# Deletion of Von Hippel–Lindau Interferes with Hyper Osmolality Induced Gene Expression and Induces an Unfavorable Gene Expression Pattern

**DOI:** 10.3390/cancers12020420

**Published:** 2020-02-12

**Authors:** Alexander Groß, Dmitry Chernyakov, Lisa Gallwitz, Nicola Bornkessel, Bayram Edemir

**Affiliations:** Department of Medicine, Hematology and Oncology, Martin Luther University Halle-Wittenberg, Ernst-Grube-Str. 40, 06120 Halle (Saale), Germany; alexander.gross.halle@gmx.net (A.G.); Dmitry.Chernyakov@uk-halle.de (D.C.); lgallwitz@biochem.uni-kiel.de (L.G.); nicola.bornkessel@student.uni-halle.de (N.B.)

**Keywords:** von Hippel–Lindau, EMT like, hyperosmolality

## Abstract

Loss of von Hippel–Lindau (VHL) protein function can be found in more than 90% of patients with clear cell renal carcinoma (ccRCC). Mice lacking Vhl function in the kidneys have urine concentration defects due to postulated reduction of the hyperosmotic gradient. Hyperosmolality is a kidney-specific microenvironment and induces a unique gene expression pattern. This gene expression pattern is inversely regulated in patients with ccRCC with consequences for cancer-specific survival. Within this study, we tested the hypothesis if Vhl function influences the hyperosmolality induced changes in gene expression. We made use of the Clustered Regularly Interspaced Short Palindromic Repeats (CRISPR)/Cas9 technology to inhibit functional Vhl expression in murine collecting duct cell line. Loss of Vhl function induced morphological changes within the cells similar to epithelial to mesenchymal transition like phenotype. Vhl-deficient cells migrated faster and proliferated slower compared to control cells. Gene expression profiling showed significant changes in gene expression patterns in Vhl-deficient cells compared to control cells. Several genes with unfavorable outcomes showed induced and genes with favorable outcomes for patients with renal cancer reduced gene expression level. Under hyperosmotic condition, the expression of several hyperosmolality induced genes, with favorable prognostic value, was downregulated in cells that do not express functional Vhl. Taken together, this study shows that Vhl interferes with hyperosmotic signaling pathway and hyperosmolality affected pathways might represent new promising targets.

## 1. Introduction

Renal cell carcinomas (RCC) are a heterogeneous group of cancers and are among the top 10 cancers worldwide. RCC arises from renal tubular epithelial cells and more than 80% of all renal neoplasms belong to RCC [1]. The major RCC subtypes are clear cell RCC (ccRCC) with a frequency of around 70–80%, papillary RCC with a frequency of around 10%–15%, and chromophobe RCC with a frequency of around 3–5% [2]. RCC incidence increases with age and is higher for men than women. Risk factors for RCC are, for example, obesity, hypertension, cigarette smoking, chronic kidney disease, hemodialysis, renal transplantation, or acquired kidney cystic disease [3]. Moreover, genetic risk factors are involved in the pathogenesis of RCC including the von Hippel–Lindau (VHL) gene, the protein polybromo-1 gene (PBRM-1), and the SET Domain Containing 2 (SETD2) gene [4,5].

VHL is a tumor suppressor that plays a pivotal role in the development of ccRCC and gene alterations can be found in up to 90% of ccRCC cases [6]. VHL can be altered and transmitted rarely in an autosomal dominant fashion, which is associated with the VHL disease, or in most cases to a sporadic manner [6].

Several studies have been performed to generate ccRCC in mouse kidneys by inactivating Vhl. The first study used the phosphoenolpyruvate carboxykinase (Pepck)-Cre to generate proximal tubule-specific knock out (KO) mice. These mice developed a modest phenotype and after 12 months 25% of the mice had renal microcysts [7]. Using Ksp1.3-Cre, as deleter Cre, led to generation of distal tubule and collecting duct (CD) specific deletion of Vhl. These mice developed hydronephrosis but no further abnormalities [8]. However, the combined KO of Vhl together with the phosphatase and tensin homolog (Pten) resulted in hyperproliferation and kidneys with multiple epithelial tubule cysts in the cortex and medulla. A further study on mouse showed that deletion of Vhl caused increased medullary vascularization and, as a physiological consequence, developed a diabetes insipidus like phenotype by excretion of highly diluted urine. The authors hypothesized that the increased medullary vasculature alters salt uptake from the renal interstitium, resulting in a disruption of the osmotic gradient and impaired urinary concentration [9]. The rate-limiting factor in the urine concentration is the expression of the aquaporin-2 (Aqp2) water channel. Aqp2 is expressed in the principal cells of the collecting duct, and the binding of the antidiuretic hormone vasopressin (AVP) to the vasopressin type 2 receptor induces the translocation of Aqp2 bearing vesicles to the apical plasma membrane [10]. The expression of Aqp2 in the mentioned mouse model was decreased [9]. The expression of Aqp2 on the mRNA level is regulated by the cAMP-responsive element-binding protein [11] and by the action of the nuclear factor of activated T cells 5 (Nfat5) [12]. Nfat5 is activated by hyperosmotic environment of the kidney [13]. It has been recently shown that in renal cancer Nfat5 expression is targeted by microRNAs that led to reduced expression of Nfat5 target genes [14].

The cells of the renal inner medulla are challenged with a hyperosmotic environment, the driving force for water retention. We have shown that this environment is also important to regulate a specific gene expression pattern of several kidney-specific genes [15]. Further, we have recently shown that the expression of osmolality affected genes is inversely regulated in the ccRCC samples compared to normal tissue, and we were able to generate an Osm-score that allows the prediction of patients’ survival [16]. We were also able to induce the expression of the E74-like factor 5 (ELF5), a tumor suppressor in RCC [17], in the 786-0, VHL deficient, RCC cell line under hyperosmotic cell culture conditions. Interestingly, the expression level was higher in 786-0 that ectopically expressed wild type VHL [16], suggesting that VHL somehow interferes with hyperosmolality associated gene expression.

Based on these data we hypothesized that Vhl also plays an important role in the expression of hyperosmolality induced genes and that loss of Vhl function induces a ccRCC like phenotype in a normal murine collecting duct cell line. Indeed, the results of this study showed massive functional, morphological abnormalities and changes in gene expression that are Vhl and osmolality dependent.

## 2. Results

### 2.1. Generation of VHL-Deficient Cells

We have used the murine mpkCCD cells to analyze the role of Vhl in the collecting duct. This cell line has been intensively used to analyze the regulation of Aqp2 and the role of Nfat5 on hyperosmotic adaptation and they are capable of genetic manipulation [18,19]. We decided to use the CRISPR/Cas9 method to efficiently knock out functional Vhl protein expression. We used 3 different guide RNA sequences (Appendix A) and a non-targeting (Scr) sequence. Single cells were isolated and mutations within the Vhl locus were analyzed by Sanger sequencing using specific primer pairs (Appendix A). The type of mutation was analyzed by the online tool Tracking of InDels by Decomposition (TIDE) [20]. Based on this analysis, we selected single clones that showed InDels leading to a frameshift (Appendix A).

Based on these results, the functional expression of Vhl should be lost in these clones. To validate this on protein level, Western blot experiments were performed. Since the loss of Vhl stabilizes the expression of Hif1a, we have also tested if this is the case in our model. As a control, we used Scr gRNA expressing cells and 5 Vhl-targeted single-cell clones. Vhl protein expression was lost in clones H6 and G10 (Figure 1). This was associated, as expected, with Hif1a expression.

Hif1a was only detectable when Vhl protein expression was completely lost. For example, in clone D8 and C5, the expression of Vhl is weak compared to Scr. However, no stabilization of Hif1a was observed for these clones. This data shows that our cell model shows similar changes as described by other groups. In a second approach, we have analyzed the intracellular localization of Hif1a. Hif1a acts as a transcription factor and should be localized within the nucleus. We have, therefore, performed immunofluorescence analysis with Scr -cells and clone G10 using a specific Hif1a antibody. As expected, no Hif1a signal was detectable in the nucleus of Scr cells (Figure 1).

Since we were able to validate the loss of Vhl expression in these cells, we will name them as Vhl-KO hereafter.

### 2.2. Vhl Deletion Induces Loss of Epithelial Structures

Loss of Vhl is associated with an epithelial to mesenchymal transition (EMT) like phenotype [21]. We have, therefore, analyzed if this is also the case in the cell model that we used. We have performed immunofluorescence analysis using specific antibodies for markers of tight (Zona occludens 1, Zo1) and adherence junctions (β-catenin). While the control cells showed localization of ß-catenin at the cell–cell contacts, this was not the case in Vhl-KO cells (Figure 2).

Similar to β-catenin, loss of Vhl function disturbs proper tight junction assembly (Figure 2). The staining for Zo1 showed interruption of the tight junction band. This indicates that in Vhl-KO cells the epithelial cell to cell assembly is disturbed. This is also further supported by staining for the actin filaments (Figure 2). Scr cells showed actin enrichment predominantly at the cell–cell contacts, indicating an intact epithelial structure and polarity, which is not the case in the Vhl-KO cells. The cells develop a fibroblast-like phenotype with intracellular actin stress fibers and hardly any enrichment at the cell–cell contacts.

Since the changes in morphology are related to an EMT-like phenotype, we have also analyzed the mRNA expression of EMT marker genes like fibronectin, alpha smooth muscle actin, N-cadherin, and vimentin (Appendix A). We observed significant differences in expression for fibronectin and alpha smooth muscle actin. However, the expression of N-cadherin and vimentin were not affected, which could implicate an incomplete EMT.

### 2.3. Vhl Deletion is Associated with Changes in Proliferation and Migration Behavior

In the next step, we analyzed if Vhl deletion is associated with functional changes. Given that the morphological and molecular changes might represent an incomplete EMT like phenotype, we set out to determine whether these changes are associated with other phenotypic changes. We have first tested if there are differences in the proliferation rate between Scr and Vhl-KO cells using the IncuCyte S3 live-cell analysis system. We have done this by calculation of the mean doubling time of the cells. The results showed that Vhl-KO cells had significant longer doubling time, resulting in a lower proliferation rate, compared to Scr cells (Figure 3).

Since we wanted to test if Vhl function is involved in hyperosmolality affected pathways, we tested the proliferation rate of Scr and Vhl-KO cells also under hyperosmotic conditions. Hyperosmolality alone reduced the proliferation of Scr cells (Appendix A). This was also the case for the Vhl-KO cells. Under hyperosmotic conditions, however, the differences between Scr and Vhl-KO cells were still detectable. To test if the phenotype of Vhl deficient mpkCCD correlates with that of classical RCC cell lines, we tested the proliferation rate using the RCC cell line 786-0. We tested cells that do not express VHL and 786-0 cells that ectopically express human VHL (786-0-VHL). In contrast to the collecting duct cells, there were no differences between the 786-0 and 786-0-VHL expressing cells (Appendix A).

Besides cell proliferation, we have analyzed the migration behavior of Scr and Vhl-KO as well as that of the 786-0 and 786-0-VHL RCC cells by scratch wound healing assay using the IncuCyte S3 live-cell imaging system. The results showed that Vhl-KO cells migrate at a significantly faster speed (~25% faster) compared to Scr cells (Figure 4A and Appendix A). Similar to the results obtained for cell proliferation, VHL expression in 786-0 cells has a different effect on cell migration compared to the mpkCCD cells. The ectopic expression of VHL induced a significantly higher cell migration speed (Appendix A).

So far the data showed that functional deletion of Vhl in mpkCCD cells is associated with massive changes in cell morphology, proliferation, and migration. These differences are cell context-specific since 786-0 RCC cell lines showed different effects. All these experiments were performed with cells cultivated under normal (isoosmotic) cell culture conditions. Since we postulate that Vhl has an osmolality dependent function, we have repeated the analysis under hyperosmotic conditions. In contrast to proliferation, the Vhl-KO cells behaved differently in the cell migration analysis under hyperosmotic conditions. While the Vhl-KO cells migrated faster under isotonic conditions, this was reversed under hyperosmotic conditions (Figure 4B).

### 2.4. Vhl Deletion Affects Expression of Hyperosmolality Regulated Genes

These results showed that Vhl deletion has a cell and osmolality specific effect on cellular behavior. We next asked if this is also associated with changes in the gene expression level. The expression level of Aqp2 served as a marker gene. The water channel Aqp2 expression in mpkCCD cells is either induced by vasopressin stimulation or by hyperosmotic cultivation conditions. Studies have shown that the expression of Aqp2 was decreased in Vhl deficient mice. Therefore, we cultivated the Scr and Vhl-KO cells under hyperosmotic conditions and analyzed Aqp2 gene expression by real-time PCR. The expression of Aqp2 is nearly lost in Vhl-deficient cells (Appendix A). This indicates that Vhl deletion has a direct effect on AQP2 expression and probably interferes with hyperosmotic pathways. To identify additional genes that are differentially expressed in Vhl-KO cells, we cultivated Scr and Vhl-KO cells at 300 or 600 mosmol/kg, isolated total RNA, and performed gene expression profiling by RNA-Seq. In Scr cells, more than 2700 genes were differentially expressed between cells cultivated at 300 vs 600 mosmol/kg (Appendix A). For example, Ranbp3l, Prss35, or Slc6a12 are within the top upregulated genes (Appendix A). These genes were also identified in primary cultured inner medullary collecting duct (IMCD) cells [15], which indicates that the mpkCCD cell line behaves similarly to primary cultured IMCD cells. We next compared Scr cells with Vhl-KO cells cultivated at 300 or 600 mosmol/kg. The deletion of Vhl was always associated with massive changes in gene expression. The total number of differentially expressed genes was over 4700 for the 300 and more than 4200 genes for the 600 mosmol/kg comparison (Figure 5).

Functional analysis identified enrichment of genes in specific Kyoto Encyclopedia of Genes and Genomes (KEGG) pathways. Within the top 20 enriched KEGG pathways using the list of genes that were differentially expressed in the 300 mosmol/kg comparison only one cancer-associated pathway (“proteoglycans in cancer”) was detected. The top enriched KEGG pathway was “metabolic pathways” (Appendix A). Similar analyses were performed with the differentially expressed genes in cells cultivated at 600 mosmol/kg. Again, the top enriched pathway was “metabolic pathways”. In contrast to the 300 mosmol/kg comparison, more cancer-associated pathways were enriched namely “pathways in cancer”, “viral carcinogenesis”, “proteoglycans in cancer”, and “central carbon metabolism in cancer”. Two of the high-ranking pathways are “focal adhesion” and “regulation of actin cytoskeleton”, revealing higher gene expression for f-actin proteins but also actin-binding factors like vinkulin or α-actinin. Furthermore, high ranking is the “PI3K-Akt pathway” that is strongly associated with ccRCC tumors [22]. Interestingly, these data support the observed morphological and functional changes in Vhl-KO cells since these pathways are associated with cell morphology and migration.

The screening of the gene expression data for classical EMT marker genes showed that the expression of desmin is significantly induced in Vhl-KO cells. The expressions of other markers like Snail1, Snail2, Zeb1, or Axl [23] were not affected (Appendix A). Again, this might be explained by an incomplete EMT like phenotype.

### 2.5. Loss of Vhl Function Leads to an Unfavorable Gene Expression Pattern

The data of TCGA and the Human Pathology Atlas [24] allowed the identification of prognostic genes that are associated with favorable or unfavorable clinical outcome. We have, therefore, analyzed if the loss of Vhl function has an impact on expression of genes that are prognostic for patients with renal cancer. However, the Human Pathology Atlas does not discriminate between the renal cancer entities.

We have used genes that showed at least 2/−2 log_2_ fold changes in gene expression and that are prognostic on clinical outcome of the patients. About 151 genes fitted to the scheme. 91 genes were associated with unfavorable and 60 with a favorable clinical outcome (Figure 6).

When we compare the changes in expression, we observed that Vhl-KO cells showed reduced expression of 33 unfavorable and induced expression of 22 favorable genes. But the upregulated expression of more unfavorable genes (56) and predominantly reduced expression of favorable genes (38) indicates that, in summary, the loss of functional Vhl in the mpkCCD cells induces an unfavorable gene expression pattern.

We have shown that the expression of hyperosmolality induced genes is reduced in RCC samples and that a gene signature of osmolality affected genes can be used for the prediction of patient’s clinical outcome [16]. We have, therefore, analyzed if this is also the case in the present study. We have generated a list of genes that are upregulated by hyperosmolality and have a favorable prognostic outcome for patients with RCC. This list was compared with the list of genes that were differentially expressed (and a log_2_ fold change of at least 1/−1) in Vhl-KO cells under hyperosmotic conditions. We identified 51 genes that met the criteria (Figure 7). Only 5 genes were higher expressed compared to Scr in Vhl-KO cells under hyperosmotic conditions. The majority, 46 genes, were downregulated in expression. This again demonstrates that loss of Vhl induces an unfavorable gene expression pattern. These data also show that Vhl has an influence on the expression of hyperosmolality affected genes.

Vhl-KO reduces, for example, the expression of Fxyd2, Fxyd4, Rnf183, and Ranbp3l, which are prognostically favorable in patients with ccRCC [16]. Since the Human Pathology Atlas does not discriminate between the RCC entities, we have used selected genes and queried the KIRC TCGA database if they could serve as prognostic markers for patients with ccRCC. In all cases, high expression of these genes was associated with a significant overall survival of the patients (Appendix A).

Vice versa, we have also analyzed if the Vhl-KO leads to induced expression of unfavorable genes, which are downregulated by hyperosmolality. Moreover, 20 genes showed upregulation in expression and only 3 downregulated expression in Vhl-KO cells (Appendix A).

## 3. Discussion

The CRISPR/Cas9 technology has been used in a study before to delete VHL in the RENCA renal cancer cell line [25] where the authors described an EMT like phenotype due to a Vhl knock out. To our knowledge, this study is the first one that used a healthy renal epithelial cell line to introduce CRISPR/Cas9-mediated Vhl deletion and characterizes the phenotype of the cells. The limitations of the study might be: 1. that we used renal collecting duct cells, although the ccRCC is originated from proximal tubulus and 2. The use of a murine cell line. However, we are convinced that this was the right strategy to test the hypothesis that Vhl function interferes with hyperosmolality affected gene expression.

We successfully introduced mutation into the Vhl locus, leading to a frameshift and expression of nonfunctional Vhl protein. Loss of Vhl function induced stabilization of Hif1a. The deletion of Vhl was associated with loss of epithelial structure that is similar to the phenotype observed in RENCA cells [25]. Similar to RENCA cells, loss of Vhl induces a more metastatic phenotype in mpkCCD cells as they migrate faster. However, the 786-0 RCC cell line showed different behavior. Ectopic expression of VHL was associated with an increased cell migration speed. In contrast to the cell migration analysis, Vhl-KO cells showed a slower doubling time. There were no differences observed between 786-0 and 786-0-VHL cells. This indicates that Vhl deletion has a cell type-specific effect on cellular function. However, the knockdown of Vhl in lung cancer cell lines showed similar effects to what we observed in the mpkCCD cells, higher migration and lower proliferation capacity [26]. These data show that the mpkCCD cell line is a suitable model to study the role of Vhl in renal cells. Traditionally it has been thought that ccRCC originates from cells of the proximal tubulus [1]. However, there is also evidence that subsets can also originate from distal tubulus or even collecting duct [27,28,29,30]. Therefore, these studies indicate that the use of the mpkCCD cells as a collecting duct cell line might not represent a major limitation. A mouse model using Hoxb7-Cre as driver to delete Vhl expression in the collecting duct developed epithelial disruption, fibrosis, and hyperplasia [31]. However, Vhl deletion alone is not sufficient and only in combination with deletion of other genetic factors it was possible to induce ccRCC. The combined loss of Vhl, Tp53, and Rb1 induced, for example, ccRCC [32]. The same group showed that renal Vhl deletion is associated with disturbed urine concentration capability [9]. More than 14 different cell types are involved in the urine concentration and water retention in the kidneys representing a specific transcriptome [33]. Most of the water retention is mediated by the action of aquaporin water channel family [10]. The driving force for water transport is a cortico-medullary osmotic gradient. The cells of the renal medulla are faced with a hyperosmotic environment. We have also shown that the hyperosmotic environment induces a kidney and even cell-specific gene expression pattern [15]. In a recent study, we have shown that the hyperosmotic gene expression pattern is lost in ccRCC samples and that this has also consequences for patients’ outcome [16]. In the mentioned study, the initial gene list was generated in rat primary collecting duct cells [15,16] and we were able to develop a translational comparison from healthy rat cell to human renal cancer and survival prediction, showing the translational potential of the data [16].

In the collecting duct, the rate-limiting factor in water retention is the water channel Aqp2. The expression of Aqp2 is downregulated in Vhl-deficient mice [9,32]. Downregulation of Aqp2 has been also shown in patient-derived ccRCC samples [34,35]. The expression of Aqp2 is regulated by the action of Nfat5 transcription factor. Nfat5 is activated by the hyperosmotic environment [13]. The group of Schönenberger et al. postulated that Vhl deletion induces reduction of the osmotic gradient by the increased angiogenesis that could lead to decreased Aqp2 expression [9]. Loss of Aqp2 is also evident in a mice model that developed renal cancer [32]. We have shown that hyperosmolality regulates the expression of several hundred genes (including Aqp2), that this expression pattern inversely correlates with ccRCC tumor samples, and that this can be used for prediction of cancer-specific survival [15,16]. Within this study, we have shown that Vhl deletion has a direct negative effect on Aqp2 expression. Besides the massive morphological changes, loss of Vhl induced changes in gene expression. More than 4700 genes were differentially expressed compared to control cells under isosmotic conditions. Loss of Vhl function was associated with more than 8500 differentially expressed genes in the RENCA RCC cell line after CRIPSR/Cas9 deletion of Vhl [25]. Unfortunately, the total list of genes and the used significance level is not published to compare the list of genes. However, in both cases, loss of Vhl function alone induces massive changes in gene expression in either cancer or normal cell lines. KEGG Pathway analysis showed that loss of Vhl affects, for example, “PI3K-Akt signaling pathway” or “Regulation of actin cytoskeleton” pathway. Dysregulation of these pathways could explain the observed EMT like phenotype. Since we were able to detect increased expression of selected markers genes for EMT like fibronectin and smooth muscle actinin, no changes were observed for vimentin, E-cadherin, or Snail1. Therefore, the observed changes might represent an incomplete EMT. Since the expression of fibronectin is induced, the observed changes might be due to pro-fibrotic changes. Besides Col1a1 no other classical markers like Mmp9, Timp1, or Col3a1 were induced on gene expression level. As explained for EMT, this might indicate a partial or mild pro-fibrotic change. However, further studies are needed to specifically analyze if the observed changes might represent incomplete EMT, pro-fibrotic changes, or a mixture of both.

The KEGG pathway analysis might be used for the identification of novel therapeutic targets. For example, targeting PI3K-Akt pathways has been in focus in treatment of different cancer types including RCC [22,36]. These data show that Vhl deletion in mpkCCD cells induces a gene expression associated with a cancer-related phenotype and this is also supported by the enrichment of genes that are involved in “Pathways in cancer”. This is also supported by the comparison of our data with the data from the Human Pathology Atlas [24]. Vhl deletion induced the expression of more genes that are unfavorable and predominantly reduced expression of genes that are associated with a favorable outcome of patients with RCC. Nonetheless, the data from the Human Pathology Atlas does not discriminate between the RCC entities. However, the query of the TCGA KIRC cohort for selected genes showed that they could serve as prognostic markers for patients with ccRCC.

Since the cells in the inner medulla of the kidneys are faced with a hyperosmotic environment, we also compared the gene expression pattern in Vhl-KO and Scr cells under hyperosmotic conditions. Several genes are regulated by changes in hyperosmolality and their expression is inversely regulated in ccRCC samples compared to normal tissue [15,16]. For example, the expression of the E74 like ETS transcription factor 5 is not detectable in ccRCC samples and ectopic ELF5 expression reduced tumor development in mice [17]. This indicates that ELF5 can act as a tumor suppressor in ccRCC. We have shown that Elf5 expression is highly induced by hyperosmotic environment [15]. The expression of ELF5 was also inducible in the 786-0 ccRCC cell line when the cells were cultivated under hyperosmotic conditions. However, the level of induction was even more striking in 786-0-VHL cells [16]. When we compare the influence of Vhl on hyperosmolality affected gene expression again several thousand genes are affected. KEGG pathway analysis showed that 4 cancer-associated pathways are within the top 20 enriched pathways compared to one in the isosmotic comparison. There are also several genes within the significantly downregulated genes that are known to be induced in expression by hyper osmolality. For example, the expression of gamma subunit of Na K-ATPase (Fxyd2) and the FXYD domain containing ion transport regulator 4 (Fxyd4) are upregulated by hyperosmolality [15,37]. The expressions of both genes are downregulated in human ccRCC and a mouse ccRCC model [32,35]. Other prominent genes are the ran-binding protein 3 like (RanBP3L) or the ring finger 183 (Rnf183). The expression of both genes is induced by hyperosmolality [15,38,39]. However, in Vhl-KO cells, the expression of both genes is significantly downregulated. FXYD2, RanBP3L, and Rnf183 expression level is associated with clinical outcome of patients with ccRCC [16]. This is also the case in the Vhl-KO cells. About 51 genes that are induced by hyperosmolality and have a favorable outcome for patients with RCC are differentially expressed in Vhl-KO vs. Scr cells under hyperosmotic conditions and 46 of them are downregulated in expression when Vhl function is missing. We have shown that the expression of hyperosmolality affected genes are inversely regulated in ccRCC samples and that high expression of genes in patients, which are downregulated by hyperosmolality, have an unfavorable outcome [16]. These data show that Vhl-KO cells under hyperosmotic conditions express a gene expression pattern that is described to be unfavorable for patients with ccRCC. These data also support our hypothesis that loss of Vhl function is associated with disturbed hyperosmotic adaptation capacity, as it is shown by differences in proliferation and migration capacity. Of course, we have to be aware of the fact that up to 30% of patients with ccRCC have functional VHL protein or at least no genetic alterations that would affect the functional expression of VHL.

The main transcription factor that is activated under hyperosmotic condition is Nfat5 [40], inducing the expression of osmoprotective genes [41]. As an example, NFAT5 induces the expression of the solute carrier SLC6a12 [41] and the expression of Slc6a12 is massively downregulated in Vhl-KO cells (log_2_ fold change of −3.3). This implicates that probably loss of Vhl function has an influence on Nfat5 activity. Nfat5 has also a function in the immune system for the macrophage and T lymphocyte function [42], and haploinsufficiency is associated with immunodeficiency [43]. However, the role of Nfat5 in cancer is controversial. Nfat5 deficiency promoted hepatocellular carcinogenesis and metastasis [44] in one study. Another study showed that Nfat5 promoted apoptosis and inhibited invasion in hepatocellular carcinoma cell lines [45]. A further study reports that S100a4 protein promoted proliferation and migration of ccRCC cell line through NFAT5 [46]. A recent study showed that NFAT5 is a target of metabolically active micro RNAs (miRNA) [14]. The gene expression of NFAT5 is downregulated in ccRCC samples compared to normal tumor samples [14] as well as expression of NFAT5 target genes. The authors postulate that the miR-106b-5p and miR-122-5p are involved in the downregulation of NFAT5 and also downstream of NFAT5 target genes [14]. However, VHL-dependent regulation of NFAT5 remained unclear. Our data indicates at least an interaction of VHL and NFAT5 functions and further analysis are needed to identify the molecular mechanisms in more detail. Since Vhl deletion alone did not induce renal cancer in mice model, it might be interesting to test if a Vhl/Nfat5 double KO develops renal cancer.

Taken together, we have shown that Vhl deletion in collecting duct cells induces an EMT like phenotype, an unfavorable gene expression pattern, and that loss of Vhl function significantly regulates the expression of hyperosmolality expressed genes that are favorable prognostic markers for patients with ccRCC.

## 4. Material and Methods

### 4.1. Cell Culture

HEK293T cells were obtained from the DSMZ-German Collection of Microorganisms and Cell Cultures and cultivated in Dulbecco’s Modified Eagle’s Medium (DMEM) supplemented with 10% serum (fetal calf serum (FCS)) and 1% penicillin/streptavidin. The 786-0 and ectopic VHL expressing 786-0-VHL cells were a kind gift of Prof. Barbara Seliger [47]. These cells were maintained in DMEM supplemented with 10% fetal calf serum (FCS), 2 mM glutamine, 1 mM pyruvate, and 1% penicillin/streptomycin. The mpkCCD cell line was a kind gift of Prof. Mark Knepper [48]. These cells were cultivated in DMEM Ham F-12 medium supplemented with 10% FCS and 1% penicillin/streptavidin. All cell lines were cultured at 37 °C and 5% CO_2_. The medium osmolality was adjusted to 600 mosmol/kg by the addition of 100 mM NaCl and 100 mM urea to the corresponding medium.

### 4.2. Oligos and Primers

The selection of the sequences for the guide RNAs (gRNA) targeting Vhl was performed according to the DNA 2.0 online tool [49] Three different sequences were selected (Appendix A). Real-time PCR primers and the PCR primers for amplification of the targeted Vhl locus were designed by NCBI Primer BLAST [50]. All oligos were purchased from Biolegio B.V. (Nijmegen, The Netherlands).

### 4.3. Cloning of gRNAs and Vector Production in Escherichia coli 

Three different gRNAs targeting murine Vhl locus and random scrambled (Scr) gRNA were cloned into lentiCRISPRv2. The lentiCRISPR v2 was a gift from Feng Zhang (Addgene plasmid # 52,961; Addgene, Watertown, MA, USA). The cloning was performed as described [49]. Plasmid isolation was performed using GeneJET Miniprep Kit (Thermo Scientific, Waltham, MA, USA). Isolated plasmids were analyzed by Sanger sequencing (Eurofins Genomics, Ebersberg, Germany) using human U6 primer. Positive plasmids were used for further experiments.

### 4.4. Vectors

The lentiviral particles were produced in HEK293T cells. For the production, the pLP1, pLP2, and pLP/VSVG vectors were used from ViraPower™ Lentiviral Packaging Mix (Thermo Fisher, Waltham, MA, USA). The HEK293T cells were cultivated to 70% confluency. Cell culture medium was reduced to starve cells and then lentiviral vectors pLP1 (7.2 µg), pLP2 (2.4 µg), pLP/VSVG (4.0 µg), and ligated lentiCRISPRv2 vector (10.4 µg) were added to the medium together with transfection reagent Turbofect™ (Thermo Fisher). Cells were incubated for 24 h. After exchange, medium cells were incubated for another 48 h. The conditioned virus-containing medium was removed, sterile filtrated, and kept at −20 °C.

### 4.5. Viral Transfection of mpkCCD Cells and Sequencing of Vhl Locus

The mpkCCD cell line was seeded into 6-well cell culture dishes and cultivated to 40–50% confluency. Medium was removed and conditioned medium containing the virus particles and fresh medium in a 1:1 ratio was added. After 48 h, medium containing 2 µg/mL puromycin for selection of transduced cells was added. Genomic DNA was isolated and targeted genomic regions were amplified by PCR using specific primers. The PCR products were purified using GenElute™ PCR Clean-up Kit (Sigma Aldrich, St. Louis, MO, USA) and analyzed by Sanger sequencing using the sequencing service of Eurofins Genomics GmbH (Eberberg, Germany).

### 4.6. Isolation of Single-Cell Clones and Tracking of Indels (TIDE)

Single-cell clones were isolated in 96-well cell culture plates by serial dilution. Total DNA was isolated and the targeting region was amplified by PCR and analyzed by Sanger sequencing. The sequences were analyzed by TIDE for the identification of specific mutations of Vhl gene [20]. Single clones harboring frameshift mutations on both alleles were finally selected and used for further analysis.

### 4.7. Western Blot

Total protein was isolated from cells using Pierce^®^ RIPA lysis and extraction buffer with protease inhibitor mix (40 µL/mL, Thermo Fisher Scientific, Waltham, MA, USA). Protein lysates were separated by SDS-PAGE with 4–12% Novex™ Bis-Tris gradient gel (Thermo Fisher Scientific). Separated protein bands were then blotted onto 0.45 µm nitrocellulose membrane by Western blotting. Successful blotting was confirmed by Ponceau S staining. All washing steps of the membrane were performed with phosphate buffered saline + 0.1% Tween 20 (PBST). Unspecific binding sites were blocked with 5% BSA or milk solution. First and horse reddish peroxidase coupled secondary antibody were applied in milk or BSA solution and incubated for 1h at room temperature. Afterward, the membrane was incubated with ECL Western Blotting Substrate (Thermo Scientific) and signals were detected on the ChemiDoc Imager detecting system (BioRad, Hercules, CA, USA). The antibody directed against Vhl (sc-55506) was purchased from Santa Cruz Biotechnology, Inc. (Dallas, TX, USA). Against Hif1a (36169S) and Gapdh (2118S) from Cell Signaling Technology, Inc.( Danvers, MA, USA).

### 4.8. Immunofluorescence

Immunofluorescence was performed as described before [15]. Cells were seeded in 24-well cell culture plates on glass cover slips and cultured to desired confluency. Medium was then removed and cells were fixed in 4% formalin. Unspecific binding sites were blocked by incubation with fishskin-gelatine (0.3% in PBS). First antibody was applied in gelatine solution and incubated at 37 °C for 1 h. Three wash steps (15 min) were performed with PBS and the cells were incubated for 1 h with the secondary Alexa-labeled antibody solution in PBS. The cells were washed three times with PBS (15 min) and mounted on glass slides. Images were taken on a Keyence BZ-8100E microscope (Keyence Corporation, Osaka, Japan).

### 4.9. Proliferation and Migration

For proliferation and migration assays, cells were cultivated either at 300 or 600 mosmol/kg in 96-well cell culture dishes. For migration assays, the wells were grown to 100% confluency. Data were collected with IncuCyte^®^ Live-Cell Analysis System (Essen BioScience, Inc., Ann Arbor, MI, USA) for 24 h. For migration analysis, a wound to the cell layer was created with WoundMaker™ (Essen BioScience) Migration capacity was evaluated by relative wound density (RWD) (Essen BioScience). Cell proliferation was also measured by live-cell imaging using the IncuCyte^®^ Live-Cell Analysis System. In a 96-well plate 1000–2000 cells were cultivated either at 300 or 600 mosmol/kg and monitored for 48 h. The doubling time was calculated by using nonlinear regression analysis and exponential growth quotation with GraphPad Prism version 5.0 (GraphPad Software Company, San Diego, CA, USA).

### 4.10. Real-Time PCR (RT-PCR)

Real-time PCR was performed as described before [15]. The expression of *Aqp2* was acquired by real-time PCR using a specific primer pair (forward: 5′CAC CGG CTG CTC CAT GAA TCC3′, reverse: 5′TCC GCC TCC AGG CCC TTG AGC3′). As reference gene, *Gapdh* was used (forward: 5′TGG CCT TCC GTG TTC CTA CC3′, reverse: 5′GGT CCT CAG TGT AGC CCA AGA TG3′). Data acquisition was done with Bio-Rad CFX Manager 3.1 Software and quantified by 2^−∆∆*C*T^ method as described [51].

### 4.11. Preparation of Samples for Next-Generation Sequencing

For gene expression analysis using Next-Generation Sequencing RNA-Seq, the cells were cultivated for 5 days to 70–85% confluency at 300 and 600 mosmol/kg. Total RNA was isolated and reverse transcribed. Within these samples, the single clone specific *Vhl* mutations were confirmed via TIDE as described above. RNA samples from 3 independent separate isolations were used for analysis by RNA-Seq by Novogene Co, Ltd. (Cambridge, UK). The quality control, sequencing, and bioinformatics were performed by Novogene as service. The detailed description can be found in Appendix A.

### 4.12. Favorable and Unfavorable Gene Expression in Kidney Cancer

A total of 2755 favorable and 3213 unfavorable genes for kidney cancer from the Human Protein Atlas Database were used. These two groups of genes were compared separately to differential expressed genes with a log_2_ fold change ≥2.0/<−2.0 between Vhl-KO and Scr cells at 300 mosmol/kg. Overlapping genes between the groups were selected and the changes in expression are displayed in waterfall plots.

### 4.13. Induced Gene Expression by Osmolality in Kidney Cancer

All induced genes with a log_2_ fold change ≥1.0 from the comparison of Scr 600 mosmol/kg (600) vs. 300 mosmol/kg (300) were compared to identify favorable genes from Human Protein Atlas database. The expression of these genes was compared with the list of differentially expressed genes (with a log_2_ fold change ≥1.0/<−1.0) between Vhl-KO vs. Scr cultivated at 600 mosmol/kg. The overlapping genes with level of expression changes are displayed as a waterfall plot.

### 4.14. Statistics

Results are expressed as the mean ± standard error of mean (SEM). Statistical evaluation was performed using GraphPad Prism 5.0 software. Comparisons were analyzed using one-way analysis of variance (ANOVA) or by two-tailed Student’s *t* test. The data were considered significant if *p*-values ≤ 0.05.

## 5. Conclusions

Our study demonstrates that loss of Vhl function alone induces massive morphological and functional changes in a healthy renal cell line. It induces an unfavorable gene expression pattern under isosmotic cell culture conditions. Vhl deletion massively interferes with the hyperosmotic gene expression program. It induces predominantly downregulation of hyperosmolality induced genes, which are predicted with favorable clinical outcome of patients with ccRCC.

Thus, targeting osmolality represents a novel promising therapeutic option for ccRCC therapy.

## Figures and Tables

**Figure 1 cancers-12-00420-f001:**
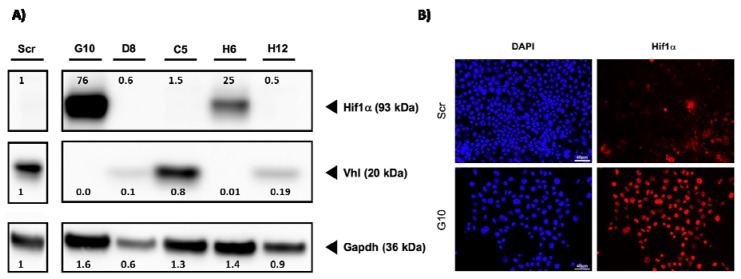
Loss of von Hippel–Lindau (VHL) protein induces nuclear Hif1A expression. (**A**) Cell lysates from control cells (Scr) and 5 VHL clones were prepared and the expression of VHL and Hif1a was analyzed by Western blot. An antibody directed against Gapdh served as control. The numbers indicate ratios in signal intensity compared to Scr. (**B**) Cells were cultivated on glass coverslips. After fixation, the cells were incubated with a specific Hif1a antibody. A secondary Alexa-568 labeled antibody was used to visualize the signals. The staining of the nuclei was done by incubation with 4′,6-diamidino-2-phenylindole (DAPI) (scale bar = 40 µm).

**Figure 2 cancers-12-00420-f002:**
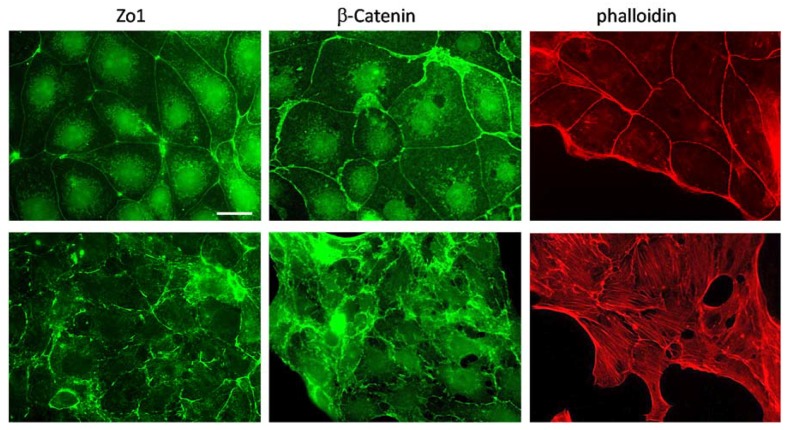
Loss of von Hippel–Lindau (Vhl) protein induces morphological changes. Cells were cultivated on glass coverslips. After fixation, the cells were incubated with specific antibodies directed against Zo1 and β-catenin. A secondary Alexa-488 labeled antibody was used to visualize the signals. For actin filament staining, after fixation, the cells were incubated with an Alexa-568 labeled phalloidin (scale bar = 20 µm).

**Figure 3 cancers-12-00420-f003:**
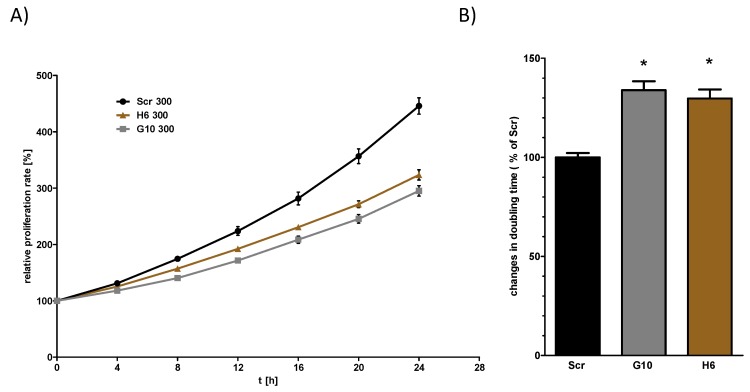
von Hippel–Lindau (Vhl) deletion is associated with longer doubling time. Cells were cultivated in 96-well plates and the proliferation was measured by live-cell imaging using IncuCyte S3 system taking an image every 4 h. The relative cell density was plotted and the doubling time was calculated by nonlinear exponential growth equation using GraphPad Prism (**A**). The doubling times were normalized to the Scr cells (**B**). One Way ANOVA was performed to identify statistically significant differences compared to Scr cell and are marked by * (*p* value < 0.05; *n* > 5).

**Figure 4 cancers-12-00420-f004:**
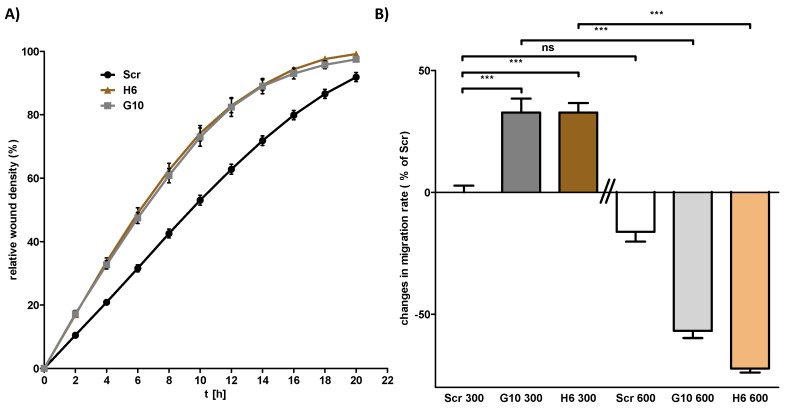
Loss of von Hippel–Lindau (Vhl) expression induces cell migration capacity. Cells were cultivated in 96-well plates until confluency and a wound to the cell monolayer was applied using the AutoScratch wound making tool. Cell migration was observed by live-cell imaging using the IncuCyte S3 system. (**A**) Representative plot of the wound density over time. (**B**) Cells were cultivated in 96-well plates until confluency either at 300 or 600 mosmol/kg. The relative wound density after 12 h was calculated by linear regression analysis using GraphPad Prism. The migration speed was normalized to Scr cells cultivated at 300 mosmol/kg. One Way ANOVA was performed to identify statistically significant differences and are marked by *** (*p* value < 0.001; *n* > 3).

**Figure 5 cancers-12-00420-f005:**
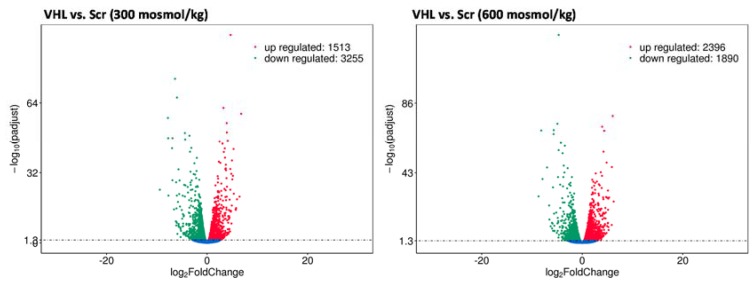
VHL (von Hippel–Lindau) deletion induces massive changes in gene expression. Scr- and Vhl-KO cells were cultivated at 300 or 600 mosmol/kg. Total RNA was isolated and gene expression was analyzed using Next-Generation Sequencing technology and differentially expressed genes were identified (*p* < 0.05, *n* = 3). The volcano plots show the number of genes, the *p*-values, and log_2_ fold changes for cells cultivated at 300 (**left**) or 600 (**right**) mosmol/kg.

**Figure 6 cancers-12-00420-f006:**
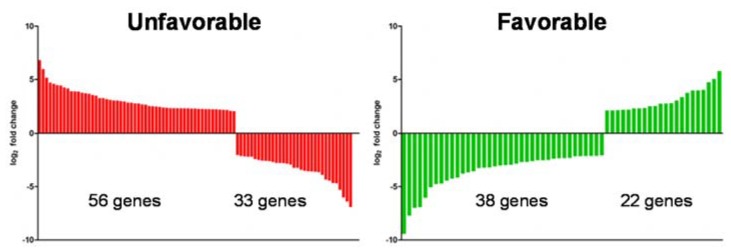
Deletion of VHL (von Hippel–Lindau) induces an unfavorable gene expression pattern. The list of genes with a log_2_ fold of 2 or higher and −2 or lower was compared with genes that have a prognostic impact on patient’s outcome with renal cell carcinomas (RCC). The left panel shows genes with unfavorable and the right with favorable prognostic outcome. The expression level after Vhl deletion is plotted as log_2_ fold change.

**Figure 7 cancers-12-00420-f007:**
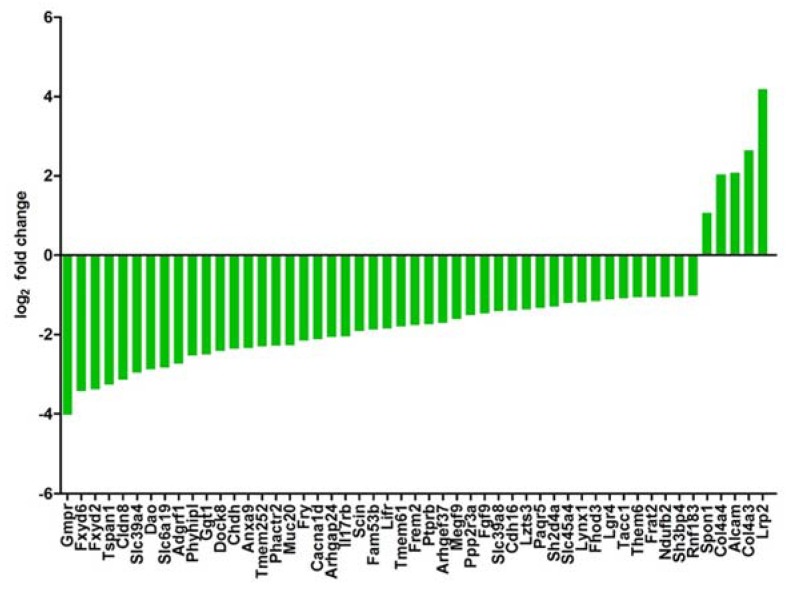
Deletion of von Hippel–Lindau (Vhl) reduces expression of hyperosmolality induced genes with favorable prognostic value. The list of genes that are (1) induced by hyperosmolality, (2) favorable for patients outcome, and (3) differentially expressed in Vhl-KO cell with a log_2_ fold change of 1 or higher and −1 or are plotted here.

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
