# Peer review of "Deletion of Von Hippel–Lindau Interferes with Hyper Osmolality Induced Gene Expression and Induces an Unfavorable Gene Expression Pattern"

_cancers, 2020, doi:10.3390/cancers12020420_

Round 1
Reviewer 1 Report
In their manuscript, the Authors describe some interesting effects of VHL deletion on the phenotype of normal renal cells. This observation might be a starting point for an interesting study; however the manuscript in its present form is still at a rather preliminary stage. More specific points are listed below:
Major:
There is no clear link between “post-EMT” phenotype of KO cells and the results of genomic analyses. If the Authors postulate that “unfavorable” changes of gene expression pattern are related to “post-EMT” phenotype, they should at least concentrate on EMT control genes/markers (vimentin/desmin/Snail/Slug etc.), and show their expression shifts upon KO; Then, they should characterize the products of these genes in the cells (localization, protein levels, effect of siRNA/chemical inhibitors/activators on cell phenotype, etc.; Accordingly, presentation of the data from transcriptomic analyses as a tool to scrutinize the mechanisms of VHL-related phenotypic shifts would be helpful. It would add much more to the impact of the paper than their discussion as the endpoints (which otherwise is rather superficial); The message on Aqp2 and its involvement in the observed phenomena is unclear; Similarly, it is unclear how the Authors link their data with the possible initiation of renal cancers; This may be important because the impact of microenvironmental factors in cancer initiation now attracts a continuously increasing attention. The Authors refer to the manuscript that apparently is still unpublished/inaccessible, which adds to the problems with grasping the idea of the story; Please, discuss mechanisms/consequences of cellular specificity of the observed reactions;
Minor:
Abstract needs to be more specific so that the reader gets the message without referring to the citation, which remains inaccessible; Introduction is (over)loaded with information of potential relevance to the study topic; however it lacks the structure, so it is difficult to follow. For example, the Authors refer to cancer development, but after going through the whole manuscript, I can’t tell whether they postulate any links between “EMT” and cancer initiation (see above); Some figures could be combined for a better clarity of the message; (Fig. 1-3, 4-5, 6-7); 5. Should be “actin” rather than “ZO-1”; 7: Please, provide the pictures showing the wound-healing; In general, the language is fine, but the are still some clumsy expressions left (“medium was achieved”, etc.)Author Response
Please see the attachment
Reviewer 2 Report
The manuscript by Alexander Gross et al. describes the generation and physiological as well as the molecular analysis of a renal collecting duct epithelial cell line where the VHL gene was inactivated by CRISPR technology.
The strength of the experimental approach lies in the effort of generating monoclonal sub-lines and verifying that both VHL alleles were successfully targeted.
The generated, clonal cell lines, in comparison to the negative control, exhibited an EMT-like phenotype and showed differences in proliferation and migratory potential.
For an translational effort, the authors performed transcriptome sequencing of the clonal cell lines under conditions of low and high osmolality. From the sequencing information, they generated gene sets that intersect with curated lists of “favorable” and “unfavorable” genes.
The experiments were well conducted with appropriate controls. My main point of concern, however, is the translational section, where a stringent line of evidence is not clearly visible. Please see specific comments below.
Major topics
The authors should more clearly specify the rationale of selecting mouse renal collecting duct epithelial cells. The majority of people place the origin of ccRCC and pRCC to proximal tubular epithelial cells. Although, as the authors pointed out, there are some reports of physiological similarities between ccRCC and collecting duct epithelial cells, the rationale of selecting these cells remains elusive.
The authors used a set of three antibodies for the definition of a EMT-like phenotype associated with the loss of VHL: Beta-catenin, Zo1 and actin. This result could be even further supported by additional using well established EMT markers. Maybe the authors could try to check for the E-/N-Cadherin switch or other makers such as ZEB1, ZEB2, Snail or Slug.
As expressed, my main point of concern is the translational section. The experimental system is a non-malignant, although SV40-transformed, murine collecting duct cell line with deleted VHL. Transcriptome sequencing data from this cell line was then used to propose gene sets of “favorable” or “unfavorable” genes for human RCC patients. Besides the species mismatch and the unconventional physiological origin of the cell line model, the translational approach has some weaknesses, that the authors should mention and thoroughly discuss:
- The curated list of “favorable” or “unfavorable” genes was derived from the protein atlas database. This database in turn relies solely on the TCGA renal cell carcinoma cohort which contains cases of clear cell (KIRC), papillary (KIRP), chromophobe (KICH) and even several Xp11 translocation cancers, that are not separately considered in the protein atlas database.
- The definition of “favorable” or “unfavorable” is based solely on the TCGA transcriptome sequencing data with median read count as separator and overall survival as the clinical end point. In the protein atlas database, only a small fraction of these genes and the proposed deregulation is experimentally validated. There is also no information present about the base expression of the gene in question. When a “favorable” gene is overexpressed, do we talk about read counts of 10 per million or 10,000 per million?
- Likewise, the TCGA cohort seems to be more like a consecutive cohort than a highly selected patient set. About 2/3 to 70% of the sporadic RCC cases have a functional VHL deletion (genomic deletion, somatic mutation, inactivation by hypermethylation………) So one should at least consider, that up to 1/3 of the patients used as a reference for comparison with a VHL deleted cell line could in fact express a functional VHL gene.
These topics are inherent to the underlying data the authors used in their translational approach and the authors can hardly overcome these problems with reasonable effort. Nevertheless, these limitations should be thoroughly discussed.
Minor topics
Fig. 5: The central column undoubtedly shows actin stress fibers but is labeled Zo1.
Fig. 6B: The y-axis label might be misleading. “Changes in proliferation rate” might lead to the impression that VHL-deleted clones proliferate faster. The authors might consider labeling this axis “doubling time”.
In Fig. 11 the authors present a set of 51 genes that have a favorable prognostic impact when expressed higher than median in RCC cases and that are induced under hyper osmotic conditions. And this induction was impaired by VHL deletion
Does a reciprocal set of genes also exist? A set of genes with un-favorable impact that are induced by hyper-osmolality?
The authors generated extensive gene lists for the intersect of genes deregulated in VHL+ vs. VHL- under low and high osmolality and the curated lists of favorable and unfavorable genes in RCC. Although this comparison has the limitations and weaknesses, as pointed out above, this gene list can nevertheless be extremely useful for leading towards new scientific approaches. The authors should definitely include the gene lists as supplementary data sets.
Please check the reference list. There are 53 numbered in-text citations but the reference list contains 54 entries.
Round 2
Reviewer 1 Report
It seems that the new version has not been considerably improved. As the Authors did not provide any responses to my comments concerning the original version of the manuscript, I could have missed some modifications. However, even these I have noticed do not clarify much the message of the story, because in my opinion the Authors failed to thoroughly discuss new data (on EMT markers). This is unfortunate, because the paper describes some interesting correlations that would potentially justify its publication in Cancers; however not in the present form. Some more specific comments are listed below:
- the Authors show the considerable up-regulation of fibronectin and smooth muscle actin; without any significant effect on the expression of other markers. This may indicate the "incomplete" EMT, which would support the leading hypothesis (this point is not discussed in the paper). On the other hand, the induction of FN/alphaSMA expression may also illustrate pro-fibrotic changes. This possibility should also be discussed, unless the Authors have the data that would falsify this notion;
- the text lacks the clarity, it also contains repetitions and clumsy expressions. I still think that some figures could easily be combined for the clarity of data presentation.
Reviewer 2 Report
The authors have adequately addressed my previous comments.
The additional experiments further support the reported physiological changes towards an EMT-like phenotype.
And with the included supplementary dataset, it is possible to recapitulate the changes in gene expression induced by osmotic conditions in a VHL-functional vs. VHL-deleted background.
Likewise, by exactly specifying the utilized database, the result section concerning the potential influence on patient survival is more stringent.
I have no additional concerns or comments
Author Response
We thank the reviewer for the comments and are happy that we were able to revise the manuscript in the suggested way.
Best regards,
Bayram Edemir
Round 3
Reviewer 1 Report
I have no more comments